# Shielding of Hepatitis B Virus-Like Nanoparticle with Poly(2-Ethyl-2-Oxazoline)

**DOI:** 10.3390/ijms20194903

**Published:** 2019-10-03

**Authors:** See Yee Fam, Chin Fei Chee, Chean Yeah Yong, Kok Lian Ho, Abdul Razak Mariatulqabtiah, Han Yih Lau, Wen Siang Tan

**Affiliations:** 1Department of Microbiology, Faculty of Biotechnology and Biomolecular Sciences, Universiti Putra Malaysia, Selangor 43400, Malaysia; cyee531@hotmail.com (S.Y.F.); yongcheanyeah@hotmail.com (C.Y.Y.); 2Nanotechnology and Catalysis Research Centre, University of Malaya, Kuala Lumpur 50603, Malaysia; cheechinfei@um.edu.my; 3Department of Pathology, Faculty of Medicine and Health Sciences, Universiti Putra Malaysia, Selangor 43400, Malaysia; klho@upm.edu.my; 4Department of Cell and Molecular Biology, Faculty of Biotechnology and Biomolecular Sciences, Universiti Putra Malaysia, Selangor 43400, Malaysia; mariatulqabtiah@upm.edu.my; 5Laboratory of Vaccines and Immunotherapeutics, Institute of Bioscience, Universiti Putra Malaysia, Selangor 43400, Malaysia; 6Biotechnology and Nanotechnology Research Centre, Malaysian Agricultural Research and Development Institute (MARDI), Persiaran MARDI-UPM, Serdang 43400, Malaysia; hylau@mardi.gov.my

**Keywords:** virus-like particle, polymer, conjugation, antigenicity, poly(2-ethyl-2-oxazoline), hepatitis B virus capsid

## Abstract

Virus-like nanoparticles (VLNPs) have been studied extensively as nanocarriers for targeted drug delivery to cancer cells. However, VLNPs have intrinsic drawbacks, in particular, potential antigenicity and immunogenicity, which hamper their clinical applications. Thus, they can be eliminated easily and rapidly by host immune systems, rendering these nanoparticles ineffective for drug delivery. The aim of this study was to reduce the antigenicity of hepatitis B core antigen (HBcAg) VLNPs by shielding them with a hydrophilic polymer, poly(2-ethyl-2-oxazoline) (PEtOx). In the present study, an amine-functionalized PEtOx (PEtOx-NH_2_) was synthesized using the living cationic ring-opening polymerization (CROP) technique and covalently conjugated to HBcAg VLNPs via carboxyl groups. The PEtOx-conjugated HBcAg (PEtOx-HBcAg) VLNPs were characterized with dynamic light scattering and UV-visible spectroscopy. The colloidal stability study indicated that both HBcAg and PEtOx-HBcAg VLNPs maintained their particle size in Tris-buffered saline (TBS) at human body temperature (37 °C) for at least five days. Enzyme-linked immunosorbent assays (ELISA) demonstrated that the antigenicity of PEtOx-HBcAg VLNPs reduced significantly as compared with unconjugated HBcAg VLNPs. This novel conjugation approach provides a general platform for resolving the antigenicity of VLNPs, enabling them to be developed into a variety of nanovehicles for targeted drug delivery.

## 1. Introduction

Over the past decades, virus-like nanoparticles (VLNPs) have received considerable interest in nanotechnology owing to their biocompatible and biodegradable properties as well as their distinct interfaces for functionalizations [1]. The rapid advances of VLNPs in recent years have led to a proliferation of studies on targeted drug delivery as they provide numerous advantages over synthetic nanomaterials [2]. VLNPs are widely employed as smart drug delivery systems by packaging and delivering therapeutic cargo such as chemotherapeutic drugs, peptides, and oligonucleotide to cancer cells [3,4,5,6]. Despite the remarkable features of VLNPs, applications of these nanoparticles as nanocarriers have some drawbacks, including high antigenicity and immunogenicity that lead to the rapid clearance of these nanoparticles mediated by phagocytes and dendritic cells. Phagocytic cells mark the opsonized nanoparticles for rapid uptake and elimination upon binding of immunoglobulin to nanoparticles, compromising drug delivery efficacy to tumor cells [7].

Recombinant hepatitis B core antigen (HBcAg) produced in *Escherichia coli* self-assembles into VLNPs containing 180 or 240 HBcAg subunits arranged into a triangulation number *T* = 3 (30 nm in diameter) or *T* = 4 (34 nm in diameter) icosahedral symmetry, respectively [8]. A variety of cargos including DNA, RNA, peptides, green fluorescent protein, and chemotherapeutic drugs have been loaded into HBcAg VLNPs for biomedical applications [3,4,5,6,9,10,11,12]. In addition, the large surface area of HBcAg VLNPs exposes a series of amino acid residues with specific functional groups (eg. Asp, Glu, Lys, and Cys) readily available for modifications through bioconjugation and chemical cross-linking. Hence, various targeting moieties and drugs can be conjugated to HBcAg VLNPs to attain targeted drug delivery for cancer therapeutics [4,5,6,13].

Despite their potential applications in targeted drug delivery, HBcAg VLNPs are highly antigenic, and they were reported to behave as T-cell-independent and T-cell-dependent antigens [14]. The configuration of the HBcAg capsid spikes protruding from the surface of the capsid may serve as a recognition site for B cell membrane receptors (BCR) [15], owing to the presence of dominant B cell epitopes at the tip of the spikes [16]. Furthermore, it has been reported that B cells rather than non-B cell antigen-presenting cells (APCs) such as macrophages and dendritic cells act as the primary APCs for HBcAg, which explains its enhanced immunogenicity in terms of antibody production [15]. Many studies in the past decades focused on the development of efficient targeted drug delivery systems using VLNPs to improve cancer therapeutic efficacy. Nevertheless, scant attention has been given to the intrinsic antigenicity and immunogenicity of the nanoparticles in drug delivery.

In the present study, we aimed to reduce the antigenicity of HBcAg VLNPs by shielding their surface with a hydrophilic biodegradable polymer, poly(2-ethyl-2-oxazoline) (PEtOx). Amine-end functionalized PEtOx (PEtOx-NH_2_) was chemically synthesized through the cationic ring-opening polymerization (CROP) technique and conjugated to the protruding spikes of HBcAg VLNPs via carboxyl groups. The resulting PEtOx-conjugated HBcAg (PEtOx-HBcAg) VLNPs were characterized with UV-visible spectroscopy and dynamic light scattering (DLS). The colloidal stability of the VLNPs was studied by incubating PEtOx-HBcAg VLNPs in Tris-buffered saline (TBS) at 37 °C for five days. The antigenicity of PEtOx-HBcAg VLNPs was then evaluated with enzyme-linked immunosorbent assays (ELISA). 

## 2. Results

### 2.1. Synthesis of PEtOx-NH_2_

PEtOx-NH_2_ was synthesized via the CROP of 2-ethyl-2-oxazoline (EtOx) monomer using an initiator, methyl *p*-toluenesulfonate (MeOTs). The reaction was performed with a monomer to initiator molar ratio of 25:1. The addition of 1,2-ethylene diamine quenched the cationic living chain ends of the PEtOx, resulting in primary-amine termination of the polymer (Scheme 1).

The proton nuclear magnetic resonance (^1^H NMR) analysis depicted in Figure 1 confirmed the chemical structure of purified PEtOx-NH_2_. As shown in the NMR spectrum, the signals at δ 1.127 and 2.776 ppm are in accordance with the –CH_3_ group from the side-chain and initiator residues, respectively. The signals at 2.305–2.410 and 3.452–3.476 ppm correspond to the –CH_2_ group from the side-chain and the ethylene imine backbone, respectively. Meanwhile, the signal at 2.704 ppm corresponds to the –NHCH_2_CH_2_-NH_2_ terminal group.

Figure 2 shows that the mass signals exhibited a major polymer distribution with a regular separation of 99 Da, which is equivalent to the molecular weight of an EtOx repeating unit.

### 2.2. Conjugation of PEtOx-NH_2_ to HBcAg VLNPs

The conjugation of PEtOx-NH_2_ to HBcAg VLNPs was performed via a two-step carbodiimide method. The carboxyl groups (Asp and Glu) of HBcAg VLNPs were covalently conjugated to the primary amine group of PEtOx-NH_2_ using 1-ethyl-3-(3-dimethylaminopropyl) carbodiimide hydrochloride (EDC) and *N*-hydroxysulfosuccinimide (Sulfo-NHS). The PEtOx-conjugated HBcAg (PEtOx-HBcAg) VLNPs were purified and their wavelength absorbance, ranging from 240 to 600 nm, was measured spectrophotometrically (Figure 3). In comparison to unconjugated HBcAg VLNPs, PEtOx-HBcAg VLNPs exhibited a significantly higher absorbance at 310 nm, which corresponded to the highest absorbance exhibited by PEtOx-NH_2_ alone. Thus, this result indicates that the difference in absorbance spectra was due to the conjugation of PEtOx-NH_2_ to HBcAg VLNPs.

### 2.3. Dynamic Light Scattering (DLS) and Zeta Potential of HBcAg VLNPs

The hydrodynamic diameter of HBcAg and PEtOx-HBcAg VLNPs was 34.47 ± 0.23 and 35.55 ± 1.95 nm, respectively (Table 1). The polydispersity index of HBcAg and PEtOx-HBcAg was 0.1 and 0.05, respectively, indicating a homogenous sample population. The zeta potential value of HBcAg VLNPs was −38.40 ± 2.76 mV. Conjugation of PEtOx-NH_2_ decreased the amount of carboxylate groups displayed on the nanoparticles surface, thereby increasing the zeta potential of PEtOx-HBcAg VLNPs significantly to −31.00 ± 1.71 mV (Table 1). 

### 2.4. Colloidal Stability of HBcAg and PEtOx-HBcAg VLNPs

The colloidal stability of HBcAg and PEtOx-HBcAg VLNPs was evaluated by incubating the nanoparticles in TBS at 37 °C for five days. Measurements of the particle size were performed each day using DLS, and the results showed that both the HBcAg and PEtOx-HBcAg VLNPs maintained their size in TBS at 37 °C throughout the five days (Figure 4). Both VLNPs have a comparable particle size with no significant change in size as compared with the initial particle size at day 0. Thus, the results demonstrated that both HBcAg and PEtOx-HBcAg VLNPs were stable in TBS for at least five days at 37 °C.

### 2.5. Antigenicity of PEtOx-Conjugated HBcAg VLNPs

The antigenicity of PEtOx-HBcAg VLNPs was analyzed with ELISA. Skim milk (SM) served as a negative control while HBcAg VLNPs (HBcAg) served as a positive control. The shielding effect of different amounts of PEtOx coating was studied by conjugating different molar ratios of PEtOx-NH_2_ to HBcAg VLNPs: PEtOx-HBcAg (5:1, 10:1 and 15:1). Figure 5 shows that without the PEtOx conjugation, the anti-HBcAg antibody reacted strongly with the nanoparticles. When the protein concentrations increased, the absorbance also increased proportionally. By contrast, the antigenicity of PEtOx-HBcAg VLNPs reduced significantly for all protein concentrations tested, regardless of the increment of PEtOx coated on HBcAg VLNPs, from molar ratios 5:1 to 15:1. No significant difference was observed when more PEtOx was introduced, indicating a sufficient shielding effect at 5:1 molar ratio of PEtOx:HBcAg.

### 2.6. Comparison of the Antigenicity of PEtOx-Conjugated HBcAg VLNPs and PEGylated HBcAg VLNPs

Methoxypolyethylene glycol amine (mPEG-NH_2_) was conjugated to HBcAg VLNPs via the EDC and Sulfo-NHS coupling method. The antigenicity of mPEG-HBcAg VLNPs and PEtOx-HBcAg VLNPs was analyzed with ELISA. Skim milk (SM) served as a negative control while HBcAg VLNPs (HBcAg) served as a positive control. As shown in Figure 6, the antigenicity of HBcAg VLNPs increased significantly as the protein concentrations increased, indicating that the anti-HBcAg antibody reacted strongly with the nanoparticles. Conversely, the antigenicity of mPEG-HBcAg and PEtOx-HBcAg VLNPs reduced significantly at all the protein concentrations tested as compared with HBcAg VLNPs. Interestingly, no significant difference was observed between the shielding effect of mPEG-HBcAg and PEtOx-HBcAg VLNPs against antibody recognition.

## 3. Discussion

The intrinsic antigenicity of nanoparticles remains a major challenge in the development of efficient drug-delivery devices because the binding of unprotected nanoparticles to opsonin proteins will mark the particles for phagocytic clearance [17]. Therefore, it is of utmost importance to obtain an in-depth understanding of the interaction between nanoparticle surfaces and the complex biological environment that affects nanoparticle recognition and clearance by the immune system. Using the camouflaging technique on nanoparticles, the blood circulation half-life of nanomaterials can be improved by escaping the recognition and clearance by the mononuclear phagocyte system (MPS) [17,18,19]. This can be achieved by grafting a stealth coating layer onto the nanoparticle surface, restricting the interactions between nanoparticles and opsonin proteins that mediate phagocytic clearance [20,21,22].

Grafting the gold standard poly(ethylene glycol) (PEG) on the surface of nanoparticles has received widespread interest due to their stealth barrier that reduces protein adsorption and increases blood circulation half-life of some nanoparticles [23,24,25]. However, many studies reported that some PEGylated products can generate anti-PEG antibodies, which contradicts the claim that PEG is non-immunogenic. In addition, an unexpected immunogenic response commonly referred as the “accelerated blood clearance” (ABC) phenomenon is correlated with the rapid clearance of the repeated administrations of PEGylated nanoparticles [26,27,28,29]. Interestingly, the presence of anti-PEG antibodies in normal individuals who have never been treated with PEGylated therapeutics has also been reported [30,31,32]. This innate presence of anti-PEG antibodies in humans could compromise PEGylated therapeutics by priming the immune system against subsequently administered PEGylated substances [33,34].

Poly(2-oxazoline) (POx) is a promising hydrophilic polymer that has been known for half a century [35]. Nonetheless, this polymer has gained an increasing interest in recent years as it is highly functionalized while exhibiting a stealth property [36]. The increasing concern regarding the use of PEG and its potential immunogenicity has led to a renewed interest towards POx in both industry and academia [37,38]. In particular, POx is a non-toxic polymer with similar stealth behaviors as PEG. POx offers advantageous properties such as thermo-responsiveness, low viscosity, and high stability with a range of end-group and side-chain functionalizations [39,40]. POxylation, the coating of POx on the surface of VLNPs, could extend the circulation time of the VLNPs in vivo due to their hydrophilicity properties. It was reported that the POxylated nanoparticles were better shielded from antibody recognition in comparison to PEGylated nanoparticles due to a higher degree of polymer coating [41]. Furthermore, several in vivo studies showed that POx is non-immunogenic even after repeated intravenous and subcutaneous injections, rendering POx as a potential alternative polymer stealth coating for drug delivery [42,43]. In addition to nanoparticle conjugation, POx can also be conjugated with peptides, proteins, and drugs for polymer therapeutics [44,45,46]. This is owed to its high functionalization possibilities in which both the alpha and omega termini of POx can be functionalized by a selection of initiators and terminating agents using the CROP synthesis technique [47].

In this study, poly(2-ethyl-2-oxazoline) with a terminal primary amine group, PEtOx-NH_2_, was successfully synthesized using the CROP technique. Mass spectrometry analysis of the PEtOx-NH_2_ revealed a major mass distribution with Δ *m*/*z* = 99 Da mass spacing, which is consistent with the monomer unit mass of EtOx = 99.13 g/mol [43,48]. Using the EDC and Sulfo-NHS coupling method, the purified PEtOx-NH_2_ was covalently conjugated to the protruding spikes on the surface of HBcAg VLNPs via carboxyl groups (Scheme 2). Conjugation of PEtOx-NH_2_ decreased the amount of carboxylate groups exposed on HBcAg VLNPs, thus the zeta potential of PEtOx-HBcAg VLNPs increased significantly. In addition, the colloidal stability study demonstrated that HBcAg and PEtOx-HBcAg VLNPs were stable in TBS at human body temperature (37 °C) over a period of five days.

ELISA analysis revealed that conjugation of PEtOx to HBcAg VLNPs significantly reduced the antigenicity of the nanoparticles, indicating that the external surface of HBcAg VLNPs had been shielded by PEtOx and exhibits a stealth behavior that restrains the binding of the antibody to the nanoparticles. Chapman et al. [36] reported that the stealth effect of PEtOx is attributed to the absence of hydrogen bond donors in the polymer. Furthermore, the hydrophilicity of the polymer chain inhibits the binding of opsonin proteins, which tend to interact with hydrophobic and charged molecules for elimination [49,50,51]. It is thought that the hydrophilic polymer chains exhibit an extended pattern owing to their flexible characteristics in solution. Thus, when opsonin proteins are bound onto the surface of the polymer, the compression of opsonin proteins towards the extended polymer chains occur, thereby resulting in a higher energy conformational change. This will then give rise to an opposing repulsive force that can completely suppress the attractive force between opsonin proteins and the polymer surface [52]. 

The shielding effectiveness of PEtOx in reducing the antigenicity of HBcAg VLNPs was also compared with PEGylated nanoparticles. ELISA analysis demonstrated that both PEtOx and mPEG significantly reduced the antigenicity of HBcAg VLNPs. No significant difference was observed between the shielding effects of these two polymers in reducing the antigenicity of HBcAg VLNPs. This indicates that both PEtOx and mPEG offer a comparable shielding effectiveness for the nanoparticles against antibody recognition. 

Despite the reduced antigenicity of PEtOx-HBcAg VLNPs, the mass distribution of PEtOx-NH_2_ might result in heterogeneous polymers binding to HBcAg VLNPs. In addition, a certain layer of polymer thickness is essential to suppress opsonin proteins effectively. Yet, the physical parameters of polymers such as the molecular weight and surface density also play a significant role in controlling the binding of opsonin proteins [52]. Therefore, further studies are required to ascertain the detailed mechanism underlying the stealth characteristics that diminish the interactions between nanoparticles and opsonin proteins. Moreover, these PEtOx-conjugated VLNPs can be further attached with other drugs or cancer-targeting ligands for specific delivery to cancer cells. Therefore, the minimal topological difference (MTD) method is crucial in studying the interactions between the ligands displayed on VLNPs and cancer receptors [53]. In addition, animal study can also be conducted to determine the immunogenicity and half-life of PEtOx-HBcAg VLNPs in blood-circulation.

## 4. Materials and Methods

### 4.1. Expression and Purification of HBcAg VLNPs

As described by Tan et al. [54], HBcAg (residues 3–148) was produced in *E. coli* strain (W3110IQ) (Edinburgh, UK) harboring pR1-11E plasmid (Edinburgh, UK). The cells were harvested by centrifugation at 8000× *g* for 30 min at 4 °C and washed with Tris-triton buffer [50 mM Tris (pH 8.0), 0.1% (*v*/*v*) Triton X-100]. After sonication, the cell extract was recovered by centrifugation at 14,000× *g* for 20 min at 4 °C. Then, the cell lysate was precipitated with 35% (*w*/*v*) ammonium sulfate, and the pellet was collected by centrifugation. The pellet was dissolved in Tris-NaCl buffer [50 mM Tris (pH 8.0), 100 mM NaCl] and dialyzed against the same buffer (1 L, two times) at 4 °C. The dialyzed protein samples were applied on a 8–40% (*w*/*v*) sucrose density gradient and fractionated by ultracentrifugation at 210,000× *g* (SW 41 Ti rotor, Beckman Coulter, Brea, CA, USA) for 5 h at 4 °C [55]. Fractions containing HBcAg as analyzed by sodium dodecyl sulfate (SDS) polyacrylamide gel electrophoresis (PAGE) were pooled and dialyzed against Tris-NaCl buffer at 4 °C. The concentration of the purified HBcAg VLNPs was determined by the Bradford assay, whereas their purity was confirmed by SDS-PAGE.

### 4.2. Synthesis of PEtOx-NH_2_

PEtOx-NH_2_ was synthesized by the cationic ring-opening polymerization (CROP) technique. A flame-dried borosilicate vial was charged with MeOTs initiator (0.5 mmol, 0.093 g, 75 μL), EtOx monomer (12.5 mmol, 1.24 g, 1.26 mL), and acetonitrile (2.5 mL) with a total monomer to initiator molar ratio of 25:1, as described by Karadag et al. [48]. The vial was then capped and heated at 80 °C for 20 h with stirring. To quench the polymer chains, ethylenediamine (5 mmol, 0.3 g, 0.33 mL) was added, and the termination reaction was performed at 45 °C for 72 h [56]. After the reaction mixture was cooled to room temperature, it was transferred to ice-cold diethyl ether followed by filtration. By using a regenerated cellulose dialysis membrane with 1000 Da cut-off (Cole Parmer, Vernon Hills, IL, USA), the yellowish polymer mixture was dialyzed against distilled water for 24 h at room temperature. The polymer solution was then freeze-fried, and the purified PEtOx-NH_2_ was recovered in 83% yield. 

### 4.3. Characterization of PEtOx-NH_2_ with NMR and Mass Spectrometry

NMR spectra were acquired at 20 °C using the Bruker Avance III HD (600 MHz) NMR spectrometer (Billerica, MA, USA). All measurements were recorded in CDCl_3_ solution. Chemical shifts were reported in ppm relative to CDCl_3_. Data for ^1^H NMR are reported as follows: chemical shift (ppm) and multiplicity (br = broad, s = singlet, m = multiplet). Mass spectrometric analysis was performed on the Agilent 6500 series accurate mass Q-TOF (Santa Clara, CA, USA). The sample (5 μL) was injected into an Agilent Zorbax Eclipse Plus C18 (4.6 × 100 mm, 3.5 μm) column with a mobile phase consisting 100% (*v*/*v*) acetonitrile. An isocratic elution was performed at a flow rate of 0.3 mL/min for 20 min. Nitrogen was used as the sheath gas. The capillary temperature and the voltage were set at 250 °C and 3 kV, respectively. The data were analyzed using the Agilent MassHunter Qualitative Analysis B.05.00 software.

### 4.4. Conjugation of PEtOx-NH_2_ to HBcAg VLNPs

Conjugation of PEtOx-NH_2_ to HBcAg VLNPs was performed via the EDC and Sulfo-NHS coupling method. The carboxylate groups of HBcAg VLNPs were activated by incubating HBcAg VLNPs (0.04 mM) with EDC (0.6 mM) and Sulfo-NHS (0.6 mM) in sodium phosphate buffer (25 mM NaH_2_PO_4_/Na_2_HPO_4_, pH 7.0; 300 μL) at 4 °C for 2 h. Then, PEtOx-NH_2_ (0.2 mM) was added into the reaction mixture containing the activated HBcAg VLNPs. The mixture was incubated at 4 °C for 20 h and dialyzed extensively against sodium phosphate buffer at 4 °C using 10 kDa cut-off dialysis membranes (Sigma-Aldrich, St. Louis, MO, USA) to eliminate excess PEtOx-NH_2_, EDC, and Sulfo-NHS. 

### 4.5. UV-Visible Spectroscopy

UV-visible measurement of HBcAg VLNPs (0.7 mg/mL), PEtOx (0.7 mg/mL), and PEtOx-HBcAg VLNPs (0.7 mg/mL) was determined using a spectrophotometer (Jenway 7315, Staffordshire, UK). The absorbance at wavelengths ranging from 240 to 600 nm was measured at room temperature.

### 4.6. Dynamic Light Scattering (DLS) and Zeta Potential Measurement

The protein samples were prepared at a concentration of 0.25 mg/mL in sodium phosphate buffer (25 mM NaH_2_PO_4_/Na_2_HPO_4_, pH 7.0). The samples were filtered using 0.2 μm syringe filter membranes and loaded into folded capillary zeta cells (DTS1070, Malvern Instruments, Worcestershire, UK). The hydrodynamic radii (R_h_) and surface charge characteristics of the HBcAg VLNPs and PEtOx-HBcAg VLNPs were analyzed at 25 °C with a Zetasizer Nano ZS (Malvern Instruments, Worcestershire, UK) equipped with a 633 nm laser.

### 4.7. Colloidal Stability of HBcAg and PEtOx-HBcAg VLNPs

The colloidal stability of HBcAg VLNPs and PEtOx-HBcAg VLNPs was studied by incubating the nanoparticles in TBS [50 mM Tris-HCl (pH 7.6), 150 mM NaCl]. The protein samples were prepared at a concentration of 0.25 mg/mL in TBS and incubated at 37 °C over a period of five days. The samples were filtered using 0.2 μm syringe filter membranes and loaded into folded capillary zeta cells. The particle size of HBcAg VLNPs and PEtOx-HBcAg VLNPs at different time intervals (0, 1, 2, 3, 4, and 5 days) was analyzed at 25 °C with a Zetasizer Nano ZS.

### 4.8. Antigenicity of PEtOx-HBcAg VLNPs

The antigenicity of PEtOx-HBcAg VLNPs with PEtOx to HBcAg molar ratios of 5:1, 10:1, and 15:1 was determined using ELISA. The purified PEtOx-HBcAg and HBcAg VLNPs were diluted to 2, 1, and 0.5 μg/mL with sodium bicarbonate buffer (50 mM, pH 9.6). The samples (100 μL) were loaded in triplicate into microtiter plate wells followed by incubation at 4 °C for 16 h. The wells were then rinsed with TBS [50 mM Tris-HCl (pH 7.6), 150 mM NaCl] containing 0.05% (*v*/*v*) Tween 20. Milk diluents (1:20 dilution; 200 μL; KPL, Gaithersburg, MD, USA) were added to each well and incubated at room temperature for 2 h. After that, the wells were rinsed with TBS-Tween buffer and incubated with the mouse anti-HBcAg monoclonal antibody C1-5 (1:1500 dilution; 100 μL; Santa Cruz Biotechnology, Dallas, TX, USA) for 1 h. The wells were then rinsed with TBS-Tween buffer followed by addition of anti-mouse antibody (1:5000 dilution; 100 μL; KPL, Gaithersburg, MD, USA), and incubated for another 2 h. Then, the wells were rinsed again with TBS-Tween buffer. The substrate *p*-Nitrophenyl phosphate (*p*-NPP; 100 μL) was added into each well, and the microtiter plate was incubated for 20 min at room temperature for color development. The absorbance at 405 nm was determined with the ELx800™ Absorbance Microplate Reader (BioTek Instruments, Winooski, VT, USA).

### 4.9. Comparison of the Antigenicity of PEtOx-Conjugated HBcAg VLNPs and PEGylated HBcAg VLNPs

Amine-terminated poly(2-ethyl-2-oxazoline) (PEtOx-NH_2_, M_w_ 5000 Da) and methoxypolyethylene glycol amine (mPEG-NH_2_, M_w_ 5000 Da) were purchased from Ultroxa^®^ Sigma-Aldrich and Alfa Aesar (Lancashire, UK), respectively. The conjugation of PEtOx-NH_2_ and mPEG-NH_2_ to HBcAg VLNPs was performed via the EDC and Sulfo-NHS coupling method as described in Section 4.4. The purified PEtOx-HBcAg, mPEG-HBcAg, and HBcAg VLNPs were diluted to 2, 1, and 0.5 μg/mL with sodium bicarbonate buffer (50 mM, pH 9.6) and coated on microtiter plate wells as described in Section 4.8. Their antigenicity was determined with ELISA as described in Section 4.8. 

### 4.10. Statistical Analysis

Statistical analysis was performed with GraphPad Prism 7.04 (GraphPad Software Inc., San Diego, CA, USA). Significant differences among the antigenicity of PEtOx-HBcAg, mPEG-HBcAg, and HBcAg VLNPs were determined using two-way analysis of variance (ANOVA) and Tukey’s multiple comparisons test, where *p* < 0.05 is considered significant, *p* < 0.001 is very significant, and *p* < 0.0001 is extremely significant.

## 5. Conclusions

In summary, an amine-end functionalized poly(2-ethyl-2-oxazoline) (PEtOx-NH_2_) was synthesized with the CROP technique and characterized with NMR and mass spectrometry. The purified PEtOx was then used to shield HBcAg VLNPs via the EDC and Sulfo-NHS coupling method. Conjugation of PEtOx to HBcAg VLNPs significantly reduced the antigenicity of the nanoparticles. Hence, these VLNPs have the potential to be employed as powerful drug deliveries in nanotechnology with the ability to evade the immune surveillance.

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
