# Peer review of "Shielding of Hepatitis B Virus-Like Nanoparticle with Poly(2-Ethyl-2-Oxazoline)"

_ijms, 2019, doi:10.3390/ijms20194903_

Round 1
Reviewer 1 Report
In this manuscript, Fam et al. developed an amine-end functionalized poly(2-ethyl-2-oxazoline) polymer shield to protect virus-like nanoparticles (VLNPs) against antibody recognition. The authors thoroughly studied and confirmed the conjugation of the polymer shield onto the VLNPs. The immunogenicity of polymer-coated VLNPs were significantly reduced by the shielding effect. While this is an exciting work, there are still some critical questions that need to be thoroughly addressed (see comments below). These questions necessitate a minor revision to this manuscript before it could be considered for acceptance.
Additional Comments:
1. Results in Figure 3 is problematic. The signal from PEtOx-HBcAg was overall higher than HBcAg sample, while the shape of the curve appeared very similar. Could there be a concentration difference between these two samples? Please update the results or comment on why the 240~280 nm background signal from PEtOx-HBcAg sample was higher than the other two samples.
2. Results in Figure 4 can be further improved. It would be interesting to study the shielding effect of different amount of PEtOx coating. Could the immunogenicity of VLNPs be further reduced if more PEtOx were introduced? This study will offer more insight on the completeness of the polymer shield.
3. Comparison with established PEGylated nanoparticles should be performed. Since both PEG and PEtOx offer shielding effect against antibody recognition, it would be informative to compare their shielding effectiveness. Please add some studies to compare the new platform with the standard PEGylated nanoparticles.
4. Another important aspect worth investigating is nanoparticle stability in biological medium. Polymer coating might potentially offer better colloidal stability for the nanoparticles. Please study the nanoparticle stability in biological media such as TBS or diluted serum.
Author Response
Comments and Suggestions for Authors:
In this manuscript, Fam et al. developed an amine-end functionalized poly(2-ethyl-2-oxazoline) polymer shield to protect virus-like nanoparticles (VLNPs) against antibody recognition. The authors thoroughly studied and confirmed the conjugation of the polymer shield onto the VLNPs. The immunogenicity of polymer-coated VLNPs were significantly reduced by the shielding effect. While this is an exciting work, there are still some critical questions that need to be thoroughly addressed (see comments below). These questions necessitate a minor revision to this manuscript before it could be considered for acceptance.
Response: Many thanks for the constructive comments. Please refer to the following point-by-point responses to the comments.
Additional Comments:
1. Results in Figure 3 is problematic. The signal from PEtOx-HBcAg was overall higher than HBcAg sample, while the shape of the curve appeared very similar. Could there be a concentration difference between these two samples? Please update the results or comment on why the 240~280 nm background signal from PEtOx-HBcAg sample was higher than the other two samples.
Response: Both the HBcAg and PEtOx-HBcAg samples were of the same concentration (0.7 mg/mL) as determined using the Bradford assay. The signal of PEtOx-HBcAg at 240–280 nm was relatively higher than HBcAg sample due to the conjugation of PEtOx-NH2 to HBcAg VLNPs. When auxochrome (–NH2) from the primary amine group of PEtOx was attached to the chromophore (–COOH) from the carboxyl group of HBcAg VLNPs, the color-imparting properties of the chromophore were enhanced, resulting in a shift of the absorption bond (λmax) to a longer wavelength with increasing absorption intensities (εmax) known as the hyperchromic effect (Yadav, 2013). Thus, the absorbance of PEtOx-HBcAg at wavelength 240–280 nm was relatively higher than the other two samples. The concentration of the samples has now been added in the caption of Figure 3 (page 4, lines 113–114) and Materials and Methods (page 9, lines 292–293) of the revised manuscript.
Reference: Yadav, L.D.S. Ultraviolet (UV) and visible spectroscopy. In Organic spectroscopy. Yadav, L.D.S.,1st ed. Springer, Netherlands, 2005; pp. 7–51.
2. Results in Figure 4 can be further improved. It would be interesting to study the shielding effect of different amount of PEtOx coating. Could the immunogenicity of VLNPs be further reduced if more PEtOx were introduced? This study will offer more insight on the completeness of the polymer shield.
Response: Many thanks for the suggestion, and we have performed the experiment as suggested. The shielding effect of different amounts of PEtOx coating was studied by conjugating different molar ratios of PEtOx-NH2 to HBcAg VLNPs (PEtOx:HBcAg): 5:1, 10:1 and 15:1. ELISA analysis revealed that the antigenicity of PEtOx-HBcAg VLNPs was significantly reduced at these molar ratios. No significant difference was observed when more PEtOx was
introduced, indicating a sufficient shielding effect at 5:1 molar ratio of PEtOx:HBcAg. This information has now been added in the sections of Results (page 5, lines 136–145), Figure 5 and its caption (page 5, lines 146–150), as well as the Materials and Methods (page 9, line 310–311) of the revised manuscript.
3. Comparison with established PEGylated nanoparticles should be performed. Since both PEG and PEtOx offer shielding effect against antibody recognition, it would be informative to compare their shielding effectiveness. Please add some studies to compare the new platform with the standard PEGylated nanoparticles.
Response: Many thanks for the suggestion, and we have performed the experiment as suggested. Methoxypolyethylene glycol amine (mPEG-NH2) and amine-functionalized PEtOx (PEtOx-NH2) with molecular mass of 5 kDa were acquired commercially, and conjugated to HBcAg VLNPs. The shielding effects of mPEG-NH2 and PEtOx-NH2 against antibody recognition were studied with ELISA. The result showed that both mPEG-NH2 and PEtOx-NH2 exhibited a comparable shielding effectiveness in reducing the antigenicity of HBcAg VLNPs. This information has now been added in the sections of Results (page 6, lines 151–160), Figure 6 and its caption (page 6, lines 161–166), Discussion (page 8, lines 231–236), and Materials and Methods (page 10, lines 323–330) of the revised manuscript.
4. Another important aspect worth investigating is nanoparticle stability in biological medium. Polymer coating might potentially offer better colloidal stability for the nanoparticles. Please study the nanoparticle stability in biological media such as TBS or diluted serum.
Response: We have performed the experiment as suggested to study the colloidal stability of HBcAg and PEtOx-HBcAg VLNPs in Tris-buffered saline (TBS) to emulate physiological conditions at human body temperature (37°C) over a period of 5 days. The particles size was determined with dynamic light scattering (DLS) at different time intervals (0, 1, 2, 3, 4 and 5 days). The DLS results showed that both HBcAg and PEtOx-HBcAg VLNPs maintained their particles size throughout the experimental period. Thus, the results indicated that both VLNPs were stable at 37°C for at least 5 days. This information is presented in the sections of Results (page 4, lines 124–131), Figure 4 and its caption (page 5, lines 132–134), Discussion (page 7, lines 213–214), as well as the Materials and Methods (page 9, lines 302–308) of the revised manuscript. The information has also been included in the sections of Abstract (page 1, lines 29–31) and Introduction (page 2, lines 77–79) of the revised manuscript.
Reviewer 2 Report
In this work, the authors have reduced the antigenicity of HBcAg VLNPs by shielding thei surface with a hydrophilic biodegradable polymer PEtOx-NH2. The resulting PEtOx-conjugated HBcAg (PEtOx-HBcAg) VLNPs have been characterized by UV-visible spectroscopy and dynamic light scattering. The antigenicity of PEtOx-HBcAg VLNPs was evaluated with ELISA.
The subject is very interesting; however, the manuscript should be improved before for publication with the following minor points:
Lines 91, 104, 116 and 121: the message “Error! Reference 120 source not found” appears in the text. Please, correct it.
Lined 110-113: How many times have the authors measured the sizes of the nanoparticles? It is impossible to express the size with so precision. According to the values of polidispersity index, it is imposible to use so significant figures. Then, the size values obtained for HBcAg VLNPs and (PEtOx-HBcAg) VLNPs are the same. The authors can not conclude that that conjugation of PEtOx-NH2 to HBcAg increased the size of the PEtOx-HBcAg VLNPs.
For this reason, the sentence about DLS of Lines 174-175 must be removed.
Author Response
In this work, the authors have reduced the antigenicity of HBcAg VLNPs by shielding their surface with a hydrophilic biodegradable polymer PEtOx-NH2. The resulting PEtOx-conjugated HBcAg (PEtOx-HBcAg) VLNPs have been characterized by UV-visible spectroscopy and dynamic light scattering. The antigenicity of PEtOx-HBcAg VLNPs was evaluated with ELISA.
The subject is very interesting; however, the manuscript should be improved before for publication with the following minor points:
Lines 91, 104, 116 and 121: the message “Error! Reference 120 source not found” appears in the text. Please, correct it.
Response: Many thanks for pointing out the “Error! Reference”. We have checked the indicated lines, and they do not refer to references, but figures and a table instead. Lines 91 (for Figure 2), 104 (for Figure 3), 116 (for Table 1) and 120 (for Figure 4). We think the errors lie in the cross-reference inserted in the text. Anyway, we have removed the cross-reference, and we hope this would rectify the errors.
Lines 110-113: How many times have the authors measured the sizes of the nanoparticles? It is impossible to express the size with so precision. According to the values of polydispersity index, it is impossible to use so significant figures. Then, the size values obtained for HBcAg VLNPs and (PEtOx-HBcAg) VLNPs are the same. The authors cannot conclude that that conjugation of PEtOx-NH2 to HBcAg increased the size of the PEtOx-HBcAg VLNPs.
For this reason, the sentence about DLS of Lines 174-175 must be removed.
Response: Many thanks for the comments. The measurement of the nanoparticles’ size was performed in triplicate. We agree with the reviewer that the size of HBcAg VLNPs and PEtOx-HBcAg VLNPs are approximately the same when the standard deviations are taken into consideration. Thus, the statements about size increment of PEtOx-HBcAg VLNPs in lines 110–111 and lines 174–175 have now been removed in the revised manuscript.